



1       **Decadal Growth in Emission Load of Major Air Pollutants in Delhi**

2       **Saroj Kumar Sahu[1], Poonam Mangaraj[1], Gufran Beig[2]**

[1]Environmental Science, Department of Botany, Utkal University, Bhubaneswar, India

[2]National Institute of Advanced Studies, Indian Institute of Science, Campus, Bangalore, India

Correspondence to: Saroj Kumar Sahu (saroj.bot@utkaluniversity.ac.in); Poonam Mangaraj

(poonammangaraj92@gmail.com); Gufran Beig (beig@nias.res.in)

**Abstract**

Indian capital megacity Delhi is reeling under deteriorating air quality and control measures are
not yielding any significant changes mainly due to a poor understanding of sources of emissions,
hence priority option in mitigation planning is lacking. In this paper, we have made an attempt to
develop a spatially resolved technological high-resolution gridded (~0.4km × 0.4km) emission
inventory for eight major pollutants of the Delhi region where high-resolution activity data of all
possible major and unattended minor sources are generated by organizing a mega campaign
involving 100s of volunteers. It is for the first time that we are able to estimate the decadal
growth in emissions of various pollutants by comparing newly developed 2020 emissions with
SAFAR emissions of 2010 using the identical methodology and quantum of activity data. The
estimated annual emission for PM2.5, PM10, CO, NOx, VOC, SO2, BC and OC over Delhi-
NCR are estimated to be 123.8 Gg/yr, 243.6 Gg/yr, 799.0 Gg/yr, 488.9 Gg/yr, 730.0 Gg/yr, 425.8
Gg/yr, 33.6 Gg/yr, and 20.3 Gg/yr respectively for the year 2020. The decadal growth (2010-
2020) in PM2.5 and PM10 are found to be marginal 31% and 3% respectively. The maximum
growth is found to be in the transport sector followed by the industrial and other sectors.
Maximum decadal growth found for pollutants BC, OC and NOx is 57%, 34% and 91%
respectively. The decadal shift of sectorial emissions with changing policies is examined. The
complete dataset is available on Zenodo at https://doi.org/10.5281/zenodo.7715595 (Sahu et al.,

27      2023).

**Keywords:** Megacity, Emission Inventory, Hotspots, Air quality, Anthropogenic Emission,
Major/Minor Sources, Mitigation Strategies



## 1.    Introduction:

Clean air is a basic need for a healthy life but air pollution has emerged as a global emergency where cities are more vulnerable due to high population density. Asian mega-cities are even more polluted than before and have drawn all global attention (IPCC, 2000; Molina & Molina, 2004; Permadi et al., 2018). Air quality in Indian megacity Delhi makes headlines across the print and media with the onset of the winter months (Beig et al., 2021; India Today, 2022). Worldwide, air pollution is a widespread problem and a major contemporary public health threat. Air pollutants are treated like a modern-day curse due to their association with premature mortality and disease burden has a significant impact on low-income developing countries, especially India. Air pollution emerged as the fourth leading risk factor contributing to disease burden and early death worldwide(HEI 2019, 2020).The Global Burden of Disease (GBD) reported that ~4.9 million premature deaths across the globe occur because of air pollution (Stanaway et al., 2018; Manisalidis et al., 2020). People from any geographical region could suffer from its adverse impacts irrespective of the place of origin (Akimoto 2003). Certainly, Indian urban have emerged as one of the most adversely affected polluting places as well as global health risks (Down to Earth, 2015; GBD, 2018). 22 cities of the world's 30 most polluted cities are in India from which Delhi, the capital of India tops the ranking for consecutive years with its annual particulate matter ($PM_{2.5}$) level nearly ten times the WHO permissible limits and is intricately caught in the toxic web of air quality and health-based standards (UNEP, 2019; World Air Quality Report, 2019, 2020). This led to alarming levels of Air Quality Index (AQI) in National Capital mega-city Delhi that has dragged first ever such a large-scale media and political attention in recent years. No doubt the mega-cities have emerged as a better place to live but at the same time, it is highly diverse across the globe and are prone to degrading air quality due to elevated concentration of particulate matter (PM) (Molina et al., 2004; Beig et al, 2020, Sahu et al, 2011, 2021). Combating mega-city air pollution become a more utmost challenge due to a poor understanding of the complexity of air pollution sources and its dynamic mixture of both natural and man-made sources.

Numerous studies have constantly manifested higher rates of respiratory and cardiovascular diseases in megacities due to alarming pollution levels where the school-going students and old generation are the largely affected (Sahu et al, 2011, Mangaraj et al, 2022).



Delhi air quality gets worse during winter months are linked with stubble burning in Punjab and
Haryana (Beig et al, 2019, 2020). The government introduced Odd & Even vehicle ply on roads
to reduce the impact of emission load (Transport Department, Govt. of Delhi, 2019). However,
the impact was not significant. The blame game keeps on running from one agency (or) state to
another where each one has its independent opinion to combat the rising level of pollutants in
Delhi. Despite many initiatives from stakeholders, Delhi air has shown no sign of improvement
and has drawn the attention of global researchers. It is confirmed that Delhi air has not improved
significantly nor safe to breathe throughout the years. Understanding the complexity of pollution
sources and their magnitude in a megacity is essential for air quality study as well as regional
atmospheric chemistry and climate point of view (Li et al, 2017). However, it becomes an utmost
challenge to identify the unattended sources and their quantification precisely, due to the
diversity of contributing major/minor sources along with the complicity of technology being
used during combustion activities. The problem becomes even more complex due to the
heterogeneity of pollution sources and their temporal variation. A comprehensive high-resolution
emission inventory (EI) may solve the purpose because EIs are critical research and regulatory
tools to address the air pollution issues in many cities. Moreover, the surface emission is the
most sensitive input data chemical transport model to understand the impact of emission on
atmospheric chemistry on different scales urban to regional, national to global scale (Sahu et al.,
2011; Mangaraj et al., 2022).
There are few limited comprehensive detail studies, that focus on Delhi emission
estimation but each study has some or the other limitations. So far, many attempts from various
attempts like that of NEERI, 2010; Guttikunda and Calori, 2013 and TERI & ARAI, 2018 have
failed to get a concrete alternative to get rid of this air quality issue/problem. In order to frame
appropriate mitigation strategies to curve air pollution load in megacity Delhi, we have identified
the new emerging sources and have estimated the pollutant load from all possible major/minor
sectors responsible for the emission of various pollutants directly or indirectly. Unlike the
previous studies, the present study is unique of its kind by targeting 17organized as well as
unorganized sectors responsible directly or indirectly for changing air quality in Delhi-NCR
regions. The present findings provide a comprehensive assessment of sources of air pollutants
and their magnitude, which has shifted with changing policies in the last one decade. One of the
main objective behind developing this reliable high-resolution (~0.4km × 0.4km) gridded



emission inventory of eight major pollutants over a domain of 70km×65km covering Delhi and its adjacent NCR region for the base year 2020 (i.e. April 2019 to March 2020) is not only to frame desired mitigation strategy to combat air pollutant issue but also to understand the decadal growth of emission over same region under the flagship of SAFAR program of MoES. It will be also an integral input to air quality forecasting based modeling study to understand the regional atmospheric chemistry.

## 1.1. Source of Emission, Activity Data & Emission Factors:

Megacity Delhi (Figure 1), the capital of India, which is designated as the National Capital Territory region (NCT), is located towards the Northern part of the country straddling the Yamuna River. This megacity is stretched over an area of 1484 km$^2$ and shares border with Uttar Pradesh in East and Haryana to the rest directions. It is situated at an elevation of ~216 m above the sea level at 28.7041° N, 77.1025° E. The NCT of Delhi is divided into nine districts. The estimated population of megacity Delhi is 28.5 million making it the largest metropolitan in India. The overgrown population density of Delhi, has led to the expansion of city and increase in use of energy and fossil fuels associated with alarming levels of air pollution and health risks.

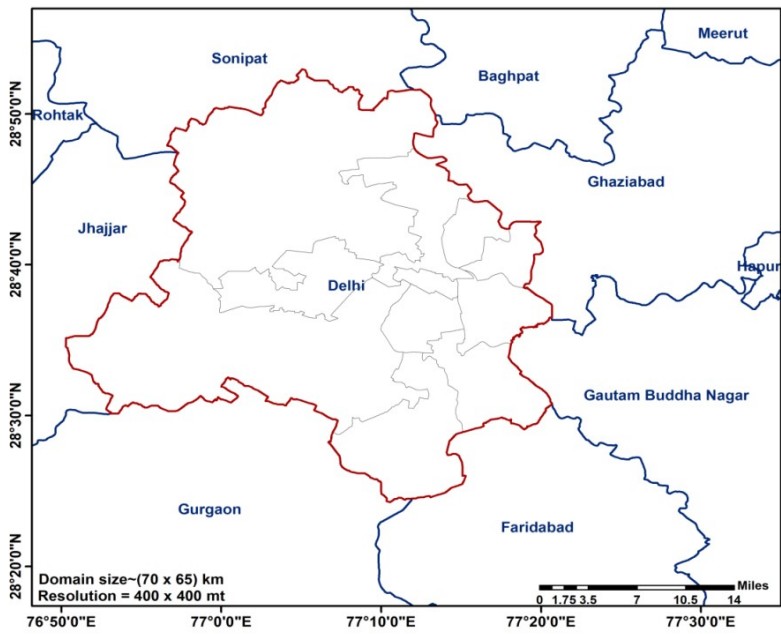

**Fig 1: Domain of interest**





In order to suffice the objective of developing an authentic emission inventory, the
collection of primary activity data is of great significance. In the present attempt, for the first
time 17minor/major sectors responsible for direct/indirect emission of pollutants have been taken
into account in the emission estimation process. To understand the emission practices, the
primary activity data were generated through a three-month-long extensive emission campaign
(SAFAR-Delhi, 2018) carried out over Delhi and surrounding National Capital Regions (NCR)
in 2018(Figure 2). This initiative was undertaken by the Indian Institute of Tropical Meteorology
(IITM, Pune) in collaboration with the School of Planning and Architecture (SPA-Delhi) and
Utkal University under the Ministry of Earth Sciences (MoES)'s project "System of Air Quality
and Weather Forecasting And Research (SAFAR)". In order to serve the purpose of
understanding the complex source of pollutants, primary activity data is of great role in building
a high-resolution gridded emission inventory, which has to be generated through a
comprehensive field campaign only. This is a unique attempt to collect micro-level primary
activity database like the type of fuel used, the quantity of fuel being in various technology in
various sectors like a slum, residential cooking, brick industry, construction sites, street vendors,
large hotels, vehicle load around tourist places/railway stations/shopping malls/large
hospitals/large school/colleges and traffic junctions, airport, biomass/crop residue burning,
crematorium, use of cow dung as an alternative fuel for cooking, road dust, construction, open
waste burning, diesel generators in commercial purpose and mobile towers. Apart from
traditionally dominating sectors like transport, wind-blown road dust, industry, thermal power
plants, and residential, there are several unattended minor sectors, which collectively have a
relatively significant contribution to air pollution issue in Delhi. Apart from this, the most
important objective is to check the authenticity and accuracy of the existing secondary data
collected from various government agencies and reports as well as to fill the data gap. For the
same, meticulously ~150 students from various universities and colleges put an extensive
painstaking approx. 40,000 hrs effort to compile a comprehensive and robust activity database
under the supervision of a group of scientists/experts. This will not only help to understand all
possible major/minor sources better but also the prevailing changing trend in megacity Delhi and
its surrounding regions. The generated data will play an instrumental role in understanding the
changing trend of the source of pollutants in the last decade.



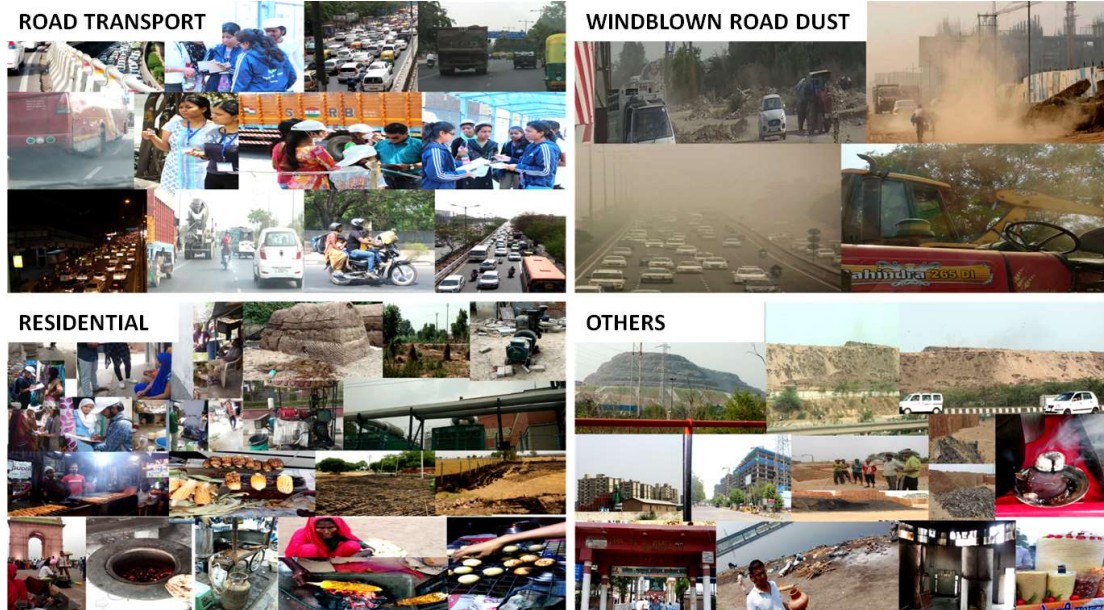


**Figure 2: Snapshot of Delhi Survey**

**(a) Transport:**
Delhi has been witnessing a consistent increase in number of motor vehicles in recent years. It is
a home to approximately 13.3 million registered vehicles as of 2020 March (MoRTH, 2020) that
has grown to two and four folds in last one and two decade respectively (Sahu et al, 2011,
SAFAR-Delhi, 2010). In transport sector, vehicles have been classified broadly into eight
categories Two-wheelers (2W), three-wheelers (3W), Buses, Personal cars, Commercial Cars,
Light commercial vehicles (LCV), Heavy commercial vehicles (HCV) and Miscellaneous
(MSLV). Overall, the relative contribution of each category showed a higher contribution of 2W
with ~56%, personal cars with ~23%, followed by commercial cars with ~17%, 3W and buses
with ~3% and the remaining 2% by rest vehicle categories. The Supreme Court of India in 1998
sanctioned a rule for all the transport system of Delhi to be run by compressed natural gas (CNG)
in order to deal with the increase in vehicular emission. Delhi as of now has ~1 million vehicles
running on CNG that constitutes ~26% of CNG-3W, 67% CNG-cars and 7% CNG-buses. The
government has been concerned for the air pollution crisis in Delhi since long and therefore BS-
IV emission norms were implemented in Delhi in 2010 before any was implemented for rest of
the nation in 2017. BS-IV has been implemented in Delhi since 2018 but has been proposed to



implement in other cities by the month April 2020. The National Automobile Scrapping policy
was introduced in India lately in 13[th] August 2021 to reduce India's vehicular air pollution with
effect from 25[th]September 2021 (MoRTH, 2021). The transport department of Delhi has lately
passed an order for diesel vehicles more than 10 years old would be deregistered automatically
from January 2022. At the same time, the calculations tally that a fraction of the fleet registered
during 2000-2010 might still be active on the roads of Delhi in 2020 despite the phasing out
process. The present area of interest has road network of ~2450 km of major roads and ~31000
km of minor roads. The manual vehicle counts were computed over 87 survey locations (Figure
3 a) in Delhi and its surrounding NCR region to identify the density of vehicle (Figure 3 b) and
its composition according to vehicle age was also estimated(Figure 3 c).
The enumerating task was carried out for both in weekdays and weekends with the help
of digital click counters. The counting task was carried out for continuously for around 14 - 16
hrs. per day. Vehicle density was recorded to be as high as around 110000 - 160000 during
weekdays in many major roads as shown in Figure 3 d. However, it was observed that vehicle
number increased during weekend over the couple of roads like India Gate Circle, Chandni
Chowk, and Lajpat Nagar etc. High vehicular density of more than 100000 per day were
observed on roads like Delhi Meerut Expressway, Dhaula Kuan, Peeragarhi, Ashram Road,
South Extension Airport Road etc. Delhi is surrounded by other populous states like Uttar
Pradesh, Haryana, and Punjab, which are directly/indirectly linked with various activities over
Delhi-NCR region. Therefore, the other state cars contribute as high as nearly 40% in majority of
well-known busy roads in Delhi. An approx. of 2600 samples was collected for the random
survey with several real time diverse data like fuel consumption pattern, hours of usage, vehicle
density, Vehicle Kilometers Travelled (VKT) per day, type of fuel used, etc. The real time VKT
generated during random survey is depicted in Table 1.




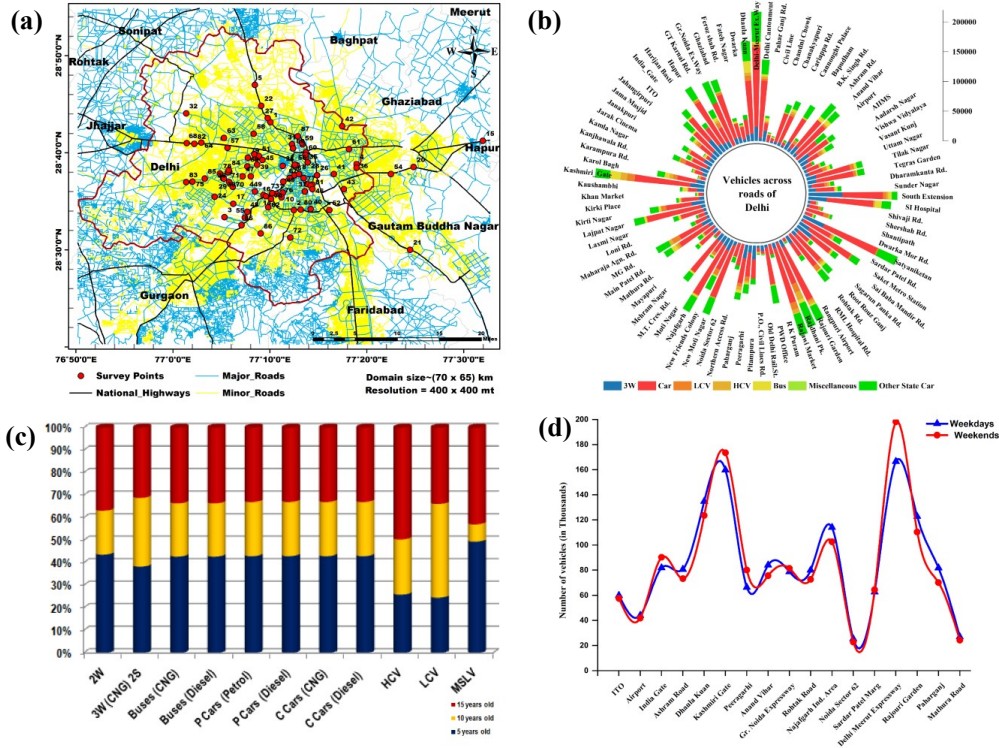

**Figure 3: (a) Survey locations for primary activity data for transport sector in Delhi-NCR; (b) Category-wise vehicle density in various roads across Delhi-NCR; (c) Age-wise vehicle category; (d) Comparison of vehicle density on weekdays and weekends on major roads of Delhi**


| Vehicle Category | Fuel | VKT (km/day) |
|---|---|---|
| Two Wheeler (2W) | Gasoline | 75 |
| Three Wheeler 2S/4S (3W) | CNG/Gasoline | 120 |
| Bus | Diesel | 210 |
| Personal Car (P Car) | Gasoline | 60 |
| Commercial Car (C Car) | CNG/Diesel | 200 |
| Heavy Commercial Vehicle (HCV) | Diesel | 75 |
| Light Commercial Vehicle (LCV) | Diesel | 150 |
| Miscellaneous (MSLV) | Diesel | 50 |

**Table 1: Vehicle Category specific VKT collected during field survey**




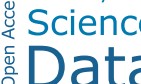

**(b) Windblown:**


Delhi has a huge and dense road network but all the roads are not certainly maintained. Road
condition of Delhi was observed keenly. The roads joining towards the outskirts of the city were
found to be worn out and lead to huge dust load. At the same time, random survey in different
roads was undertaken in order to assess the driving cycle/pattern of different vehicle categories.
The other state car contributes as high as nearly 40-50% in majority of well-known busy roads in
Delhi. Due to rise in the number of vehicles, the average speed of vehicles is found to be
decreasing in trend (i.e.18-25 km/hr in most of the major roads and 35-55 km/hr on airport roads
and few more important roads). The average weight-age of vehicles in Delhi was determined
based on vehicle category and its composition, which was estimated to be 1.23 tons. The number
of precipitation days in Delhi is hardly 50 days with an annual rainfall of just 547 mm (Rainfall
Statistics of India, 2019); therefore, the soil moisture content over study area was determined to
be considered just6%. The paved and unpaved road ratio was closely monitored and ~40% of
roads were found to be unpaved with broken road shoulders, poor infrastructure and the rest
~60% were considered to be paved. The silt load on these roads was estimated to be 10% for
paved roads and 12% unpaved roads which comparatively better than any other cities in India.
The resuspension of dust load increases with increasing weight of vehicle and speed. The
average vehicle weight and mean vehicle speed derived from fleet composition running on the
road were determined from field survey across many roads along with the number of
precipitation days and moisture content to arrive at total dust load over study area.

**(c) Industry:**


In case of industries, it is one of the most diverse sectors with more than 3182 industries
scattered over Delhi-NCR where the large fraction is much unorganized with no/limited fuel
activity data followed by small fraction of organized sector. The Central Pollution Control Board
(CPCB) and Delhi Pollution Control Committee (DPCC) have classified the polluting industries
of Delhi into three categories as Red (highly polluting), Orange (moderately polluting) and
Green (non-polluting). The red category industries are strictly banned within the Delhi city
however; orange category industries are allowed for operation. There is no comprehensive
database for all industries with their technological details. As per the primary survey, large
numbers of unorganized small industries were found to be confined over Eastern, Southeastern,





and Southwestern part of Delhi region. Central Delhi has relatively very low number of
industries in comparison to others part of city. The spatial distribution of diverse range of small,
medium and large industries is depicted in Figure 4. Major industries include – Engineering
industries, which carry a frequency of 546, Machine and tools industries of 169, Electricals 175,
Iron and Steel industries 114, etc. Most of the detailed information on industrial areas, fuel
consumption pattern, production capacity etc. has been collected from DPCC. Fuels used in these
industries include Low Sulphur heavy Stock (LSHS), Light Diesel Oil (LDO), High Speed
Diesel (HSD), Liquid Petroleum Gas (LPG), Natural Gas (NG) and coal.

### (d) Residential:

Delhi's estimated population was ~22.7 million which within a decade increased to a total
population of ~30.2 million (2020) and is known as the first most populous city of India and
second largest populated city of the world. According to the
Ministry of Housing & Urban Poverty Alleviation, Govt. of India, around 13-14% of Delhi's
population lives in slums. The Delhi Urban Shelter Improvement Board, 2019 reported ~675
clusters of slum in Delhi. The actual slum population data is very uncertain. During the field
survey, nearly 187 locations were covered to collect over 3000 samples over slum clusters
confined over the Central, Eastern and South Eastern part of Delhi. The total population is
estimated to be distributed among 4 million households with an average household size of five.
The cooking fuel activity data collected confirm the changing trend in cooking fuel used in Delhi
slum in last one decade. Unlike traditional like wood, dung, bio-fuel, LPG is being widely used
as main fuel which accounts around 95%, followed by wood 3% and coal 2%. In winters, the
relative contribution of wood as fuel increases (for heating of water). This indicates that there is
excellent penetration of government awareness and promotion of LPG connection in slum
pockets. Apart from this, it came into notice that people residing in the outskirts of Delhi are
using cow dung as fuel for heating and cooking purpose, especially during winters. The mixture
of generated agricultural residue with cow-dung and raw materials like biomass and coal dust are
still being used for domestic cooking in the peripherals of Delhi and its adjoining districts. The
mixture is dried and molded into circular shapes with a curvature staked to the walls and left for
sunbathing called as 'Uplah' in local language. Later, they are piled up into mounds to be
preserved for months and are used as an alternative for domestic fuels. As per the survey, a





single household size of 5-6 members use approx. 30kg of cow-dung per month as a source of fuel for cooking and heating of water in winter.

With changing lifestyle with eating habits, vending in megacity Delhi holds up ~5,00,000 street vendors which are well scattered all across the city. Nearly 1653 samples on cooking fuel activities were collected by interacting with people working in various hotels, restaurants and street vendors to know the exact situation prevailing in Delhi-NCR regions where the coal and wood are combusted using traditional approach as well as traditional stove. A large proportion of these street vendors were certified under the regional Municipal Corporations and were situated at permanent vending zones and many were found to be unauthorized ones who kept shifting from one place to another. During the field survey, it was observed that LPG is being predominantly used as a source of fuel by the street vendors (i.e. 83%) followed by coal (15%) and wood (2%). Few street vending zones were found to be predominantly using coal for 'tandoor' food making activities especially near tourist places like India Gate, Jam Masjid, Lal Qila etc. Kerosene is found not to be in use primarily as a source of fuel for cooking activities. However, crop residue burning is not prominent in core urban region of Delhi but the peri-urban areas towards the northeastern fringes hold less cultivated cropland. Hence, crop residue burning in the urban region is of little significance. The activity data with respect to cultivated area and amount are accounted from government portals like ICAR (Indian Council of Agricultural Research), MoSPI (Ministry of Statistics and Programme Implementation), Ministry of Agriculture & Farmers' Welfare and paid sites Indiastat. The approach used for estimating the total crop residue generated and the fraction burnt is adopted from Sahu et al., 2021.

Due to load shedding, diesel generator (DG) sets as a source for power backup are increasingly frequent in commercial establishments and apartments. In most parts of Delhi 1-2 hrs. of power failure is quiet common in summer then. Besides that, DG sets are also used in base transceiver stations (BTS). According to the Department of Telecommunication (DoT), 2019, Delhi has more than 26,000 telecom towers, which have ~1 lakh BTSs that run with DG sets for a constant or substitute source of power. A common BTS is equipped with a 12-25kWh DG set, which on an average consumes ~9000-12000 liters of diesel annually (Sahu et al., 2015). For estimating the number of diesel generator sets in commercial premises, ratio of gensets and population was taken and total number of commercial establishments in Delhi with their spatial



locations was assembled from paid sources. Total emission from DG sector was based on the
number of diesel generator sets and power failure hours.
**(e) Other:**
The Indira Gandhi International Airport is the primary international airport spread over an area
of 2066 ha situated at 9.9 miles from city centre of New Delhi. It is the busiest airport and sixth
busiest airport in Asia in terms of passenger traffic. According to the bulletin of Indira Gandhi
International Airport, the calendar year of 2019-20, it witnessed ~67 million passenger traffic
and 450,012 aircraft movements. The Landing/Take-off (LTO) cycle, which happens below the
altitude of ~ 1000 m (3000 feet) basically, contributes to the air pollution. The activity data of
aircraft movement and passenger traffic are collected from government reports of the Directorate
General of Civil Aviation (DGCA), 2020 and the Ministry of Civil Aviation, 2020.
According to the Delhi Pollution Control Committee, Delhi generates ~11,144 tons of
Municipal Solid Waste (MSW) per day on average, which are dumped over three uncontrolled
and unlined landfill sites of Delhi i.e. Ghazipur, Bhalaswa, and Okhla dump yard (DPCC, 2020).
Ghazipur landfill is the largest dump yard located towards the Eastern perimeters of Delhi
covering an area of ~70 acres, receives around 2500-3500 metric tons of solid waste every day.
Many print/news reports also stated the unexpected overflowing of wastes at the Ghazipur
landfill site. The Bhalsawa landfill is situated to the North-west of megacity covers an area of
~36 acres where every day ~2000-3000 tons of waste are dumped. The dumping ground of
Okhla is yet another landfill site with area of ~46 acres is great concern which receives at least
~1500-2000 tons of waste dumped every day despite the site was declared exhausted in 2010.
The zones covered in waste collection for Okhla includes South,Central, Najafgarh, and Delhi
Cantonment Board (DCB).
Delhi has three operational Waste to Energy (WTE) Plants of total waste intake capacity
of~5250-5750 tons per day (TPD) at three locations in Delhi namely Ghazipur, Bawana, and
Okhla. The waste-to-energy plant installed in Ghazipur has an installed capacity of 12 MW
processes ~1300-1350 TPD. The Bawana Integrated MSW plant processes ~2000-2300 TPD of
solid waste with an installed capacity of 24 MW. The Okhla WTE plant has an installed capacity
of 16 MW and processes ~1950 tons of municipal solid waste per day. Another WTE plant of

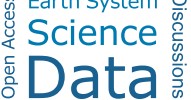

processing capacity ~2000 TPD is proposed at Tehkhand and another at GhondaGujran. After
the commission of these two proposed WTE plants, the total capacity would likely be increased
from 5750 TPD to at least 8450 TPD in upcoming future. These WTE plants potentially process
the waste for energy generation to some extent reduces the volume of landfills while providing a
renewable source of energy. Limited evidence has put forward that well-planned and well-
operated WTE plants might seem significant to reduce adverse health impacts, due to lesser
perilous emissions when compared to burning of waste at landfills, whereas, poorly fed WTE
plants potentially emit particulate matter and strenuous toxins with severe health risks (Cole-
Hunter et al., 2020). As a recent study reported that so far in India, only 23% of total generated
municipal solid waste is treated by various processing and approximately 43% of waste is
dumped some. Remaining 34% is allowed to burn openly at the landfill site itself in order to
prevent spilling over (Sharma et al., 2019). As there are three WTE plants installed within
megacity Delhi and quite evidently it processes ~22% of total MSW generated annually,
therefore it is estimated that only 48% of the total MSW is dumped and the left over 30% is
burnt right away on the dumping site which contributes to the air pollution issues in Delhi.

Along the same line, construction activities in Delhi are also one of the significant

contributors to particulate matter emission in Delhi. Construction activities include demolition,
site preparation and removal of debris. During the survey, at least 20 construction sites were
observed from which some of the major sites at DDA-Housing Sector 19-B, GH-project Sector
10, Megamall- sector 14, DDA-Housing, sector 16-B, Bhagwati C.G.H.S- Sector 22-Dwarka,
SaritaVihar- Metro enclave, Maharani Bagh flyover and Naraina flyover. HCVs and multi-utility
vehicles like bulldozers, tractors, scrapers, compactors involved in loading and unloading of
construction materials, preparation of site, demolition and disposal of debris which in a certain
way contribute to the dust load. Additional information on area and duration of construction
activities were procured from Public Works Department (PWD), 2020 and Delhi Development
Authority (DDA), 2020. In case of brick kilns industry, which is very much confined across the
outskirts of Delhi areas like Jhajjar, Faridabad and Ghaziabad region where there is a cluster of
kiln industries (like approximately 300 brick kilns in Jhajjar region). Operation of these brick
kilns is very seasonal in nature astheir peak business month between December to June month. It
is also noticed that approximately10 tons of coal or 13 tons of tudi/ rubber is being used to
produce one lakhs of bricks using semi-zig-zag technology. The sector is widely scattered in a





much unorganized manner where it is observed that coal (~70%) is being used as primary fuel
followed by tudi (i.e. mustard husk) (~25%) and rubber/other biomass/waste/etc. (~5%) as an
alternative fuel.
The practice of using Incense Sticks/Mosquito Coils/Cigarettes (IMC) has remained as an
unattended sector, which is of vital significance source to indoor and also moderately contributes
to outdoor air pollution. Use of incense sticks in festivals and holy places is common in India.
Besides that, during the field survey it was observed that maximum street vendors (this includes
both food zones and non-food zones) of Delhi as well as small scale dhabas using incense sticks
during business hours. The composition of incense sticks is responsible for continuous
smoldering. It comprises of resin, charcoal and wood dust mixed altogether and wrapped to thin
sticks made from coconut leaves or bamboos. Generally, incense sticks comprises of 45%
biomass, 25% wooden chips/bakhoor 15% coal and 15% resin/jigit (Cohen et al., 2013; Kumar et
al., 2014) and are responsible for emission of hazardous mixture of pollutants causing indoor air
pollution as well. It is very astonishing that most street vendors as well as dhaba/hotel lights
incense sticks/cake during business hours. Also, mosquito coils have been widely used by the
low/middle income grade households (Kumar et al., 2014) especially in slum zones which were
quite fascinating to observe during the field survey. The smouldering of the contents of coils:-
biomass, wood dust, and charcoal releases deadly pollutants responsible for acute respiratory
infections. Similarly, smoking of cigarettes/tobacco has caused over 10 million fatalities every
year in India. In fact, India has been declared home to at least 120 million smokers by World
Health Organization (WHO). The estimated emission for these sectors was based upon the
activity data of household population and street vendors with their per capita consumption.
Open-air funeral pyre the traditional system of cremating human bodies is a wide custom
in South Asian countries especially in India and Nepal (Chakrabarty et al., 2013) as the
population of Hindu religion is a majority. During the field campaign, around 62 crematoriums
were surveyed where it was found that only 6 crematoriums were observed to be using modern
electrical burning method as compared to 56 crematoriums with traditional method of burning of
wood. The pyre is built by using roughly ~450-550 kg of wood along with assorted materials,
such as shells of coconut, cow-dung, camphor, and pure ghee/clarified-butter. The dead body is
basically placed on top of the pyre and flaming process is carried out which takes around 4 to 6



hours. As stated by the vital statistics of Municipal Corporation of Delhi (MCD), the crude death
rate for Delhi has been reported 6.51 per 1000 people in 2020. No authentic data was accessible
regarding the number of dead bodies cremated everyday/annually in each crematorium except in
a few crematoria. So, the emission estimation was based on the population statistics on religion
data of Census from crematoriums, annual death rate, number of deaths, and quantity of wood
used. Later, the emission was spatially allocated to the respective crematorium grids. The Land
use and Land Cover pattern with activity data incorporated are highlighted in Figure 4.

**Figure 4: Land Use and Land Cover with Spatial Surrogates over Megacity Delhi and NCR**

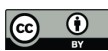



### 2.    Methodology:


Emission factors (EFs) are the most critical and sensitive components to build a reliable emission
inventory and the selection of an appropriate regional sector-specific technological emission
factor is the most crucial and challenging task and should be validated through scientific
judgments and acceptability. A dynamic EF can epitomize a better scenario of transport
emission, especially in a developing country like India where the usage of vehicle is much longer
as compared to developed countries. Based on our best judgment, in some cases, the EFs for
aging vehicle type are derived by averaging out the EFs given for 10yr& 15 yr old vehicle
category. There are few EFs which are adopted from other countries due to lack of indigenous
EFs. Although many uncertainties prevail due to the sensitivity of EFs for development of
emission inventory, but the present effort is towards the best possible estimate by including EFs
already adopted in several authenticated reports and experiments conducted by certified agencies
of government as well as government authorized non-government and autonomous agencies
which provide best estimates for EFs, that are being referred in our latest studies Mangaraj et al.,
(2022 a, b)and taken into account in this present study.

### 2.1.   Calculation:


The total emission i.e. the sum of emission from all individual sectors is expressed by Equation-1
with respect to particular pollutant. Most of sector's emission is estimated using IPCC Tire-2
approach. In the absence of activity data, the Tire-1 approach is adopted for few sectors. In case
of transport sector, the EFs are defined on the basic of kilometer travel, which is highly sensitive
to technology, and age of vehicle. In the presence of country specific technological EFs for
transport sector developed by ARAI, it is highly useful to prove the estimation. The emission
from transport sector has been calculated as per the Equation -2. In case of road dust emission,
the method is adopted from widely used AP-42, USEPA (Equation 3 and Equation 4) where the
country specific parameter like silt load, moisture content, no. of precipitation days and average
vehicular weight.

**Equations Used:**


$$TE = \sum_r \sum_s FU_{r,s}\left[\sum_t Ef_{r,s,t} A_{r,s,t}\right] \text{-----------------------------------------------------(Equation 1)}$$



Where,
r, s, t = sector, fuel type, technology, TE= Total emission, FU= Sector and fuel specific amount
Ef= Technology specific EFs, A = fraction of fuel for a sector with particular technology, where
$\sum A = 1$ for each fuel and sector.
$$E_t = \sum (Vh_l \times D_l) \times Ef_{l,km} \text{------------------------------------------------------(Equation 2)}$$
Where,
$E_t$ = Total Emission of compound, $Vh_l$ =Number of Vehicle per type, $D_l$=Distance travelled in a
year per different vehicle type, $Ef_l$, km= emission of compound, vehicle type per driven
kilometre
**For Paved Road Dust**:

$$E_p = [k\,(st/2)^{0.91}(wt)^{1.02}]\,(1 - \frac{pt}{4N}) \text{----------------------------------------------(Equation 3)}$$

Where,
$E_p$ = particulate emission factor (having units matching the units of k),k = particle size multiplier
for particle size range and units of interest, st = road surface silt loading (grams per square meter)
(g/m2),
wt = average weight (tons) of the vehicles travelling on the road, pt = number of "wet" days with
at least 0.254 mm (0.01 in) of precipitation during the averaging period, N = number of days in
the averaging period (e.g., 365 for annual, 91 for seasonal, 30 for monthly)

**For Unpaved Road Dust:**
$$E_{up} = \left\{\left[k\left(\frac{st}{12}\right)^a\left(\frac{VS}{30}\right)^d / \left(\frac{m}{0.5}\right)^c - C\right] * [(365 - pt)/365]\right\} \text{-----------------------(Equation 4)}$$

where,
$E_{up}$ = size-specific emission factor (lb/VMT), st = surface material silt content (%), m = surface
material moisture content (%), VS = mean vehicle speed (mph), C = emission factor for 1980's



vehicle fleet exhaust, brake wear and tire wear, pt = number of days in a year with at least 0.254
mm (0.01 in) of precipitation; k, a, c and d are empirical constants

**2.2.   Spatial allocation of emission:**
The Geographical Information System (GIS) organizes the geographic data from various sources
followed by being a key aspect that allows these tools to transform large spatially uniformed
emission dataset to systematic thematic layers used for developing gridded emission inventory.
A high-resolution Land Use Land Cover (LULC) digital database over the megacity is used to
improve the spatial distribution of emission from various sectors. Before input of calculated
emission into the GIS environment, several preliminary tasks like geo-referencing, digitization
and building of attribute activity database are undertaken. A GIS based statistical approach is
developed to spatially distribute the emissions across the Delhi-NCR. Different layers of spatial
proxies have been taken into account to grid the emission values to required resolution
($\sim$0.4$\times$$\sim$0.4km) for each sector, which can be used as tool for further analysis. The basic spatial
features are points, lines and polygons; layers of road networks- national and state highways,
major and minor roads; population density of village/district-level, the urban spread of the grid;
database on the economic activity of hospitals, market complexes, industrial estates, hotels,
residential blocks etc. These spatial features are used as proxies to determine the emission both
spatially and temporally where grid level emissions are allocated by overlaying the facility
location layer with the grid cell layer and aggregating the facility points in each cell covering
Delhi-NCR.
**3.     Result & Discussion:**
The developed emission inventory for major air pollutants like $PM_{2.5}$, $PM_{10}$, CO, $NO_x$, $SO_2$,
VOC, BC and OC covering Delhi-NCR in 2020 are calculated to be 123.8Gg/yr, 243.6 Gg/yr,
799.0Gg/yr, 488.9 Gg/yr, 730.0Gg/yr, 425.8 Gg/yr, 33.6 Gg/yr, and 20.3Gg/yr respectively. The
sector-wise total emission of pollutants across Delhi-NCR is provided in Table-2.Also a dataset
has been provided at https://doi.org/10.5281/zenodo.7715595 (Sahu et al., 2023) for gridded
pollutant wise sectorial spatial distribution. Keeping the space constraint in mind, comprehensive
analysis of the spatial distribution of $PM_{10}$and CO is elaborated further.



| Sector | $PM_{2.5}$ | $PM_{10}$ | CO | $NO_x$ | VOC | $SO_2$ | BC | OC |
|---|---|---|---|---|---|---|---|---|
| Windblown-Road Dust | 10.867 | 99.975 | - | - | - | - | - | - |
| Transport | 41.369 | 42.330 | 540.100 | 342.650 | 709.380 | 77.230 | 23.640 | - |
| Industry | 20.370 | 37.076 | 10.218 | 85.091 | - | 338.096 | 4.327 | - |
| Household | 0.311 | 1.310 | 1.038 | 0.867 | 0.005 | 0.227 | 0.065 | 0.113 |
| Slum | 0.216 | 0.550 | 1.443 | 0.463 | 0.010 | 0.086 | 0.018 | 0.107 |
| Street Vendor | 0.687 | 1.175 | 1.440 | 0.286 | 0.011 | 0.743 | 0.092 | 0.242 |
| Crop Residue Burning | 11.094 | 13.820 | 113.086 | 6.131 | 17.107 | 1.276 | 1.432 | 4.969 |
| Cow-Dung | 2.519 | 3.149 | 21.345 | 0.408 | 0.175 | 0.099 | 0.273 | 1.643 |
| Diesel Generators | 3.620 | 4.590 | 2.070 | 9.590 | - | 0.640 | 1.880 | - |
| Aviation | - | - | 21.007 | 36.068 | 3.297 | 2.871 | 0.021 | 0.036 |
| MSW Burning | 11.915 | 12.831 | 61.407 | 3.428 | - | 0.458 | 0.917 | 11.915 |
| WTE Plants | 10.217 | 10.441 | 0.786 | 2.021 | 0.022 | 1.853 | - | - |
| Construction | 5.956 | 9.926 | - | - | - | - | - | - |
| Brick Kiln | 2.727 | 4.017 | 12.807 | 0.913 | 0.041 | 2.106 | 0.896 | 0.773 |
| IMC | 1.161 | 1.379 | 4.143 | 0.060 | 0.005 | 0.105 | 0.031 | 0.021 |
| Crematory | 0.863 | 1.078 | 8.134 | 0.987 | 0.042 | 0.014 | 0.078 | 0.550 |
| **TOTAL** | **123.891** | **243.649** | **799.023** | **488.963** | **730.093** | **425.804** | **33.669** | **20.370** |

**Table 2:Pollutant-wise and sector-specific total emission across Delhi-NCR**

### 3.1. Anthropogenic $PM_{10}$ Emission in Delhi-NCR:

The total $PM_{10}$ emission across is estimated to be 243.6 Gg/yr, where windblown dust is emerged as largest sources (99.9 Gg/yr) followed by traditionally dominating sector like transport sector (42.3Gg/yr) and industry (37.0 Gg/yr). It is also noticed that crop residue burning (13.8 Gg/yr) and municipal solid waste burning (12.8 Gg/yr) in open area along with waste-to-energy plants (10.4 Gg/yr) are emerging as larges source of particulate matters across the city.

A high emission in the order of 1000-6000 tons/grid/yr and 120-1000tons/grid/yr is found over Central, Eastern, Northern, some parts towards the South and South-eastern fringes of Delhi confined over national highways, many major and busy roads as shown in Figure 5. Moderate emission in the order just 30-120 tons/grid/yr is well scattered across the study regions. It has been noted that Central and Eastern Delhi region are one of the highly polluted regions. Recent rising trend of vehicle numbers along with vehicle from surrounding states in Delhi road in last ten year has put tremendous pressure on road network expansion, leading heavy traffic congestion. All major traffic junctions are experiencing high emission load. However, the





highest emitting grids in the order of ~1300-6000tons/grid/yr are also found in small patches
driven by sources like WTE plants and industrial practices followed by municipal solid waste
burning as well. It has been found that the Okhla region is one of the highly polluted hotspots
where WTE plants, municipal solid waste burning followed by windblown road dust are the
dominating sectors responsible for elevated $PM_{10}$emissions.The next dominating hotspots
identified in Bawana and Ghazipur regions are dominated by large point sources like the WTE
plant with ~2566 tons/grid/yr and 1704 tons/grid/yr respectively. Furthermore, Anand Parbat
(~1300-1700 tons/grid/yr), Badli Industrial Area (~648 tons/grid/yr), Wazirpur Industrial Area
(~508 tons/grid/yr), Mayapuri Industrial Area (~400-500 tons/grid/yr), Rohini Industrial Area
(~481 tons/grid/yr) are some of the industrial dominating hotspots. It is noticed that coal is
predominantly used in both organized and unorganized industrial sector followed by diesel as
fuel. Dense major road networks across these regions led to slow-moving traffic congestion,
moreover these roads are concurrent to the major junctions of industrial area and they tend to
witness the large movement of heavy weighted HCVs and LCVs for the supply of raw materials
and goods. The continuous movement of these vehicles undoubtedly is responsible for the
broken and worn-out roads. Besides that, this vehicle-induced turbulence and poor road
condition are the leading factors accountable for road dust resuspension in an order of ~150-
750tons/grid/yr making it the second dominating sector overall. The gross weight of the HCVs
and LCVs also affects their speed while carrying the goods, which intensifies the vehicular
exhaust emission too, which is why the transport sector is the third dominating sector with ~70-
300 tons/grid/yr. High vehicular density over many busy roads is the main cause of high
particulate emission due to moderate vehicular speed of ~25 km/hr. This speed increases towards
the outskirt of the city. The load of windblown road dust depends on vehicle speed, therefore the
traffic congestion leading to a decrease in average vehicle speed in Delhi is regarded as one of
the important factors that lead to suppressing the windblown dust but at the same time, it
increases the transport emission due to traffic congestion. Apart from this, heavy commercial
vehicles loaded beyond their carrying capacity cause resuspension of road dust, which results in
severe particulate pollution. Moreover, a significant amount of vehicle fleet plying over megacity
Delhi belongs to other states where the share of the personal and commercial car (taxi) can go as
high as ~30-40% on various road types.





## 3.2.  Anthropogenic CO Emission in Delhi-NCR:

The estimated total CO emission from all the sources is found to be around 799.02 Gg/yr. The relative contributions of CO from transport, industrial, residential and other sector are estimated to be 67.5% (540 Gg/yr), 1.2% (10.21 Gg/yr), 17.5% (140.42 Gg/yr) and 13.5% (108.28 Gg/yr) respectively. The spatial pattern as shown in Figure 5 depicts that CO emission hotspots in the order of 750-6500 tons/grid/yr are found to be over the large region of Central, Eastern and South-eastern Delhi regions along with few more over surrounding NCR regions like Noida, Gurgaon, Gaziabad and Faridabad etc. Transport sector is the dominating source in the above discussed regions due to high population and dense road network driving to high vehicular activities. The estimated emission from transport is found to be around 540.10 Gg/yr, where the petrol driven vehicles emits more CO as compared to diesel and CNG vehicle. The petrol vehicles are mostly the personal vehicle in India whereas the vehicle numbers have gone up nearly two folds in Delhi during last 10 years, contribute more than 80% of total CO emission. Commercial vehicle growth contributes less to CO emission. Most of the major traffic junctions in down town are highly polluted by transports related CO emission (~1200-1800 ton/yr). Most of CO emitting Industrial zones in Delhi is more confined to Central & Eastern Delhi and few more specific regions outskirt of Delhi.

The second most dominant source is residential sector where major slum clusters contribute significantly. The regions are more confined to the Central, Eastern, South-Easter part of Delhi and few surrounding regions. It is also found that highly dense population with middle and lower income group is lying over above discuss areas too and associated slum cooking, residential cooking, street venders and commercial cooking etc. Low technological cum soil fuel based cooking practices in slum areas drive to high CO emission. Moreover, the slum population located in the Eastern and Central Delhi is dense aggravates CO emission further. A relatively low emission of the order of 25-150 ton/yr is found to be in the outskirt of Delhi and adjacent districts like Rotak, Jhajjar and Gauttam Budhanagar etc. Low population density along with agricultural lands cover are the main reasons for low emission of CO. Collectively, the street vendor cooking and commercial cooking contribute a significant amount of CO emission in densely populated regions and are well uniformly scattered over large area. Similar hotspots are

also identified over the Noida, Gurgaon, Faridabad regions surrounding the Delhi where an
emission of the order 1000-1500 tons/yr is found.

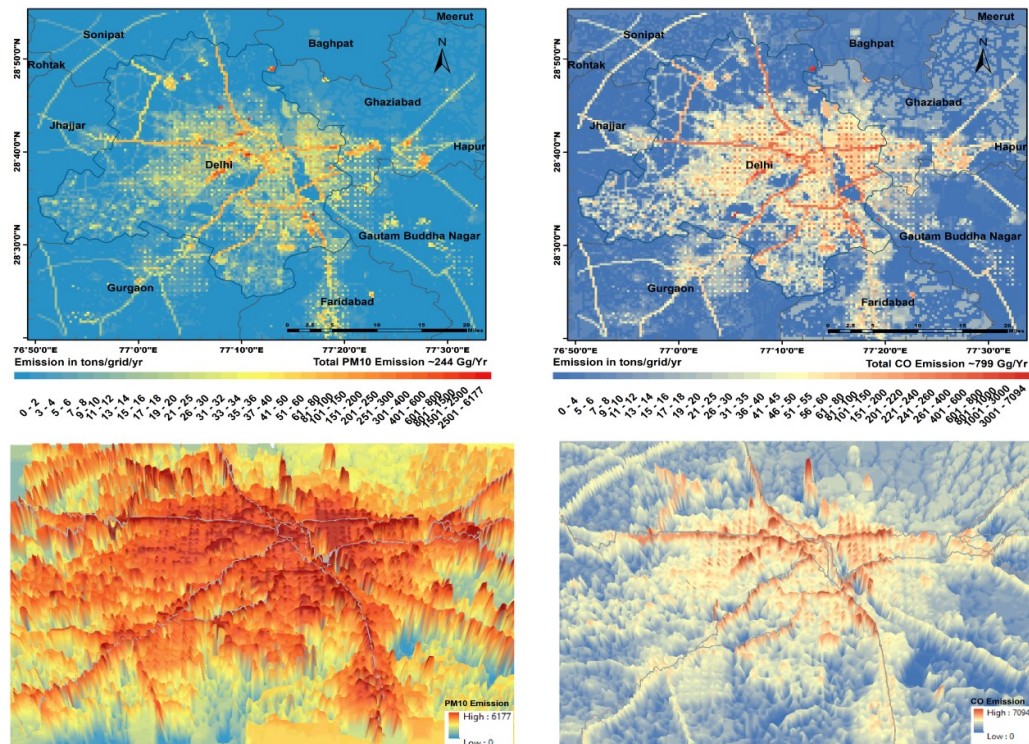

**Figure5: Spatial distribution of PM$_{10}$ and CO load across Delhi-NCR region**
**3.3.    Decadal Change in Emission (2010-2020):**
Shifting of emission sources and its trend over the years is vital to access the impact of air
pollution especially in megacities. The present estimated PM$_{10}$emission is compared with our
own previous estimation for the base year 2010 (SAFAR-Delhi, 2010) for same domain, it is
clearly concluded that the effective net increase of PM$_{10}$ emission over the last decade is just
~3%. This is small growth could be due to various new policy being adopted by government
which is directly or indirectly influence the emission. At the same time, there are couple of shift
in sectorial emission load as well as addition of new unorganized sectors in 2020 emission
estimation. If you look at the sector specific change then there are significant shift in emission
pattern and required attention. It can be observed that there has been an increase by 39% in



emission load from transport sector as compared to another36% in industrial sector during same
period. In case of windblown road dust emission, there is a decrease of 23% as shown in Figure
7. Due to penetration of LPG in slums, the cooking related emission is improved significantly as
well as in residential sector. The rise in number of vehicles with increase in spread of road
networks turned out to be the major cause along with the overburdening of four wheeler cars,
where the contribution of other state cars is significant. However, there is an increase in traffic
congestion but better paved road condition and road shoulder maintenance has resulted in a
decrease in emission load from windblown road dust in last one decade. The discontinuation
(permanent closure) of the thermal power plants in Delhi has resulted in exclusion of thermal
power plant as a sector contributing to total emission load.

As far as the residential sector is concerned, there is a rapid reduction in relative

contribution. The decrease in number of slums in Delhi when compared to 2010 period has
resulted in a reduction in consumption of cooking fuels, which shows a significant decline in
residential $PM_{10}$ load by 31%. Primarily the awareness among the people led to penetration of
LPG in slum areas, street vendors, household etc., which reduced the emissions to great extent.
However, emissions from other sectors have significant contribution to the present $PM_{10}$
load.The new emerging sectors like WTE plant, MSW burning, crematory, use of incense
sticks/mosquito coil/cigarettes and construction, were not considered in the previous report in
2010 so the relative contribution has increased significantly. This decadal change in emission is
also observed in the case of CO in similar trend except residential where there is a substantial
decreasing trend as shown in Figure6.A summary of the growth trend for all the pollutants is
shown in Table 3.





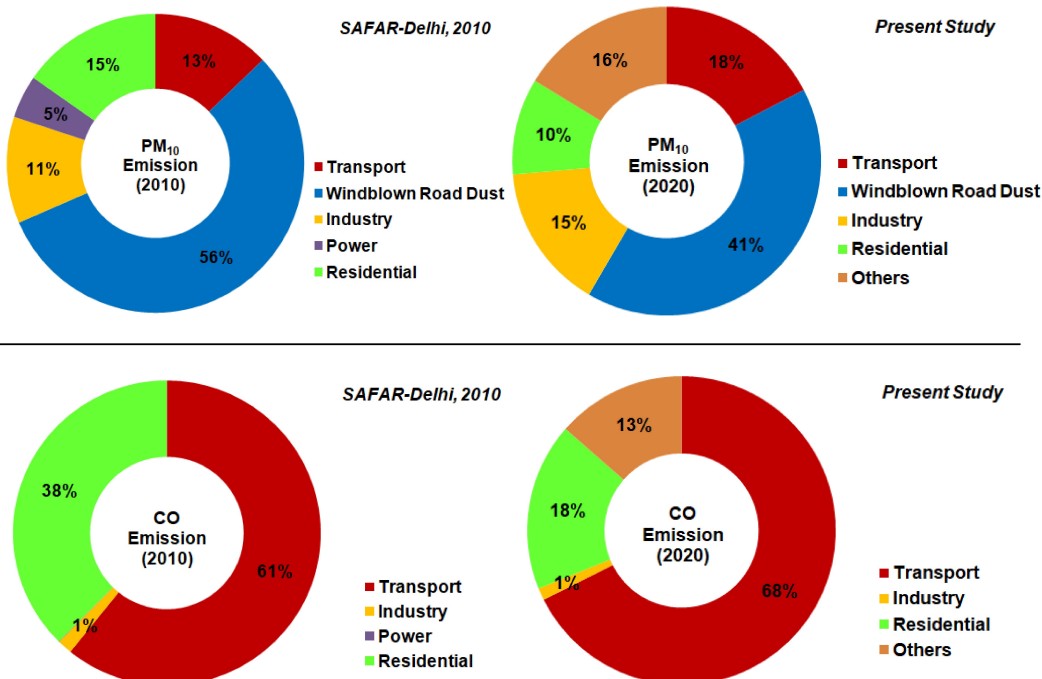


**Figure6: Decadal change of emission with sectorial relative contribution**


| Base Year | 2010* | | | | | | 2020** | | | | | |
|---|---|---|---|---|---|---|---|---|---|---|---|---|
| Sectors/Pollutants | PM$_{2.5}$ | PM$_{10}$ | CO | NO$_x$ | BC | OC | PM$_{2.5}$ | PM$_{10}$ | CO | NO$_x$ | BC | OC |
| Transport | 30.25 | 30.29 | 427.55 | 162.28 | 9.77 | - | 41.37 | 42.33 | 540.10 | 342.65 | 23.64 | - |
| Windblown Road Dust | 26.20 | 131.27 | - | - | - | - | 10.87 | 99.98 | - | - | - | - |
| Industry | 16.29 | 27.20 | 10.92 | 79.84 | 8.67 | 12.60 | 20.37 | 37.08 | 10.22 | 85.09 | 4.33 | - |
| Power | 2.87 | 11.02 | 0.29 | 6.90 | 0.04 | - | - | - | - | - | - | - |
| Residential | 18.65 | 36.07 | 264.41 | 6.40 | 2.96 | 2.60 | 18.45 | 24.60 | 140.42 | 17.75 | 3.76 | 7.07 |
| Others | - | - | - | - | - | - | 32.84 | 39.67 | 108.28 | 43.48 | 1.94 | 13.29 |
| Total | 94.26 | 235.85 | 703.17 | 255.42 | 21.44 | 15.20 | 123.89 | 243.65 | 799.02 | 488.96 | 33.67 | 20.37 |
| *SAFAR Delhi 2010; **Present Study | | | | | | | | | | | All Emission in Gg/yr | |

**Table 3: Comparison of sectorial emission during 2010 and 2020**
**3.4.    Uncertainty in emissions and limitations:**
Emission inventories may have errors due to activity data and EFs gaps. Therefore, the collection
of data and the evaluation of uncertainty are unambiguously linked. We have made an attempt to
estimate the uncertainty in the sectorial emissions for which, error propagation was calculated by



following the Monte Carlo methodology. The factors included for uncertainty estimation include
the (a) emission factors used, (b) activity data collection, (c) proxy data used, (d) data gaps
leading to approximation, and (e) efficiency of emission control. Uncertainty estimation for the
transport sector seems very complex as it involves fuel-specific technological vehicle categories
that have diversity in emission factors according to the age of vehicles. In the case of transport,
the disparity in activity data and VKT is not much as a robust ground survey was performed.
Therefore, the contribution of vehicular emission to gross uncertainty is the least with a
maximum uncertainty for ranging ±23%. Emission from windblown road dust has heterogeneous
factors like the speed of the vehicle along with its weight, soil moisture content and silt load.
These modulating factors are responsible for defining the emission load and their combined
uncertainty ranges ±33%. The residential/domestic emission source comprises of per capita fuel-
induced activity data and corresponding emission factors so the combined uncertainty in this
sector is ±28%. The industrial sector has the highest disparity in secondary activity data and its
availability of relevant technological emission factors is the key factor to a higher uncertainty
level of ±41%. The sources belonging to other sectors comprise several minor unorganized
sources, which have comparatively less contribution to total emission and have high uncertainty
ranging ±47%. The gross uncertainty in the inventory is estimated to be around ±29%, which is
found to be in an acceptable range. As of date, no comprehensive study has been done to
determine the uncertainty for the emission inventory of Delhi. This is the first approach to do the
same and in accordance with our best scientific judgment, it can be said that the present surface
emission dataset both in terms of quality and quantity has the least errors. The emission
inventory's limitation lies in various steps like limited access to industrial information like the
one fuel quantity used in various techniques used. Similarly, the exact number of other state
vehicle plying in the megacity is very uncertain and need a better approach to improve the
estimation. There are many unorganized sectors like street vendors; small-scale waste burning
across the local level, silt load on various roads, driving conditions varies with road type and its
condition etc. Still, we believe the kind of micro-level activity data used is better than any other
earlier inventories developed over the study region.

**3.5.    Inter-comparison among studies:**



In this section, a comparative analysis of the present study with the past studies is taken into
account and has been elaborated. As mentioned earlier, Delhi has been in the spotlight when air
quality issues are concern. Here, the present study is compared with previous eight studies done
over Delhi. NEERI in 2010 presented sector-wise emission inventory at 2 km resolution covering
the metropolitan area of Delhi for the base year 2007, targeting only four pollutants ($PM_{10}$, $SO_2$,
$NO_x$, CO). The calculated emissions were found to be 147 tons/day, 268 tons/day, 460 tons/day,
and 374.1 tons/day respectively. Guttikunda and Calori (2013) worked on the National Capital
Territory (NCT) region that includes Delhi and its suburbs (Gurgaon, Noida, Faridabad, and
Ghaziabad) over an area of 6400 $km^2$ at ~1 km resolution. This was done for the base year 2010
for PM, $SO_2$, $NO_x$, CO, and VOCs. It includes sectors of re-suspended road dust, construction,
vehicular exhaust, domestic cooking, power plants, industries, brick kilns, diesel gen-sets, and
waste burning. About 35% of the total $PM_{10}$ emission is contributed by the transport sector and
road dust and around 37% are contributed by the major point sources (brick kilns, industries, and
power plants). It has been highlighted that brick kilns located outside the city affect the city air to
some extent but the origin of certain sources like diesel gen-sets, waste burning, and construction
remains unclear whether they have been influenced by the surrounding areas or not. In addition,
the domain of interest considered is around 69% of the total area of Delhi, which is huge, and
therefore it doesn't give a clear representation of the exact emissions prevailing in Delhi.
Sindhwani et al., (2015) estimated $PM_{10}$, CO, $NO_x$, and $SO_2$emissions in the NCR-Delhi region
that comprises the neighboring states of Haryana and Uttar Pradesh. This study was done in the
year 2010 at a 2 km×2 km resolution. The estimated total emissions for $PM_{10}$, CO, $NO_x$, and $SO_2$
were 107.47 Gg/yr, 1290.13 Gg/yr, 342.30 Gg/yr, and 83.16 Gg/yr respectively. The contribution
of sectors like road transport, road-dust and domestic sources altogether is ~47% of total PM10
emissions. A quantitative assessment of only three pollutants i.e., PM, $NO_x$, and CO was carried
out for Delhi Urban Area for the base year 2010 by Mishra and Goyal, (2015). The major
contributors included vehicles, industries, power plants, and domestic and dust. The CO and $NO_x$
emissions from the transport sector (210.83 kt and 92 kt respectively), were found to be the
largest contributor followed by the domestic sector. Road dust (25.50 kt) has a significant
contribution to PM while vehicular, industries and power plants are approximately having equal
contributions. Similarly, Jaiprakash et al., (2016) reported an experimental-based study focusing
on specifically vehicular emissions (CO, $CO_2$, and $NO_x$) in Delhi for the base year 2012. The



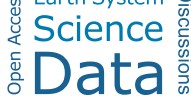

study estimates an on-road tailpipe measurement of 14 passenger cars of different types of fuel
and vintage and reported that the share of diesel, gasoline, and CNG to total CO, $CO_2$, and $NO_x$
emissions were in order of 7:84:9, 50:48:2 and 58:41:1 respectively. These studies majorly lack
in accounting for the impactful active sources like commercial cooking (street vendors),
crematoria, WTE plants, crop residue burning, and many more, which makes this inventory
insignificant for further use.
Sharma and Dikshit, (2016) attempted a comprehensive study on $PM_{10}$, $PM_{2.5}$, $NO_x$, $SO_2$,
and CO in Delhi city focusing on ~14 sources for the base year 2014 (November 2013 – June
2014) at 2 km resolution. The results showed that road dust (56%), concrete batching (10%),
industrial sources (10%) and vehicular (9%) are the major contributors to $PM_{10}$ emission.
Though the study involved site sampling for a few of the sectors it also lacks an absolute
sampling number (limitation) and most of the activity data were collected from secondary
sources. Singh et al., (2018) attempted the estimation of emissions from the road transport sector
of NCT-Delhi for the base year 2010. The study stated that major roads contribute to more than
50% of total PM emissions. When specifically focusing on limited pollutants, which most
importantly include PM, this study has certain limitations in terms of non-exhaust emission
(vehicular dust resuspension) from road transport, which is a significant contributor to the city's
$PM_{10}$ load. Thereafter, TERI & ARAI (2018) initiated a source apportionment study for
identifying sources responsible for $PM_{2.5}$ and $PM_{10}$ in Delhi-NCR and developed an coarse
resolution (4 km×4 km)based emission inventory of a few pollutants (PM, NOx, $SO_2$, CO,
NMVOC) for 2016. The results stated that in the case of $PM_{10}$, road dust and construction dust
contributed significantly, where the contribution of dust from surrounding regions was
comparatively higher in summers, which reduced the proportion of major sectors in the $PM_{10}$.
Taken as a whole, a large disparity is found between the reported past studies and present
emission estimations as shown in Figure 7. The basic reasons for these variations point towards
the differences in sectors being focused on or the activity data being considered for the past
works in conjunction with the use of technological emission factors used are also an additional
reason of concern. The base years as well as domain considered differ significantly from each
other. As the sources of emission tend to change with time and the evolution of a region hence,
upgrading an emission inventory is the most fundamental segment to be taken care of. As a

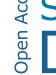



consequence, this present study has premeditated all such important factors in the most potent ways to build up this gridded surface-emission dataset. In addition to this, unlike the previous works, this study is the first-ever ultra-high-resolution-gridded (~400mts) emission data set targeting eight major pollutants for the latest base year 2020. This new dataset could be a valuable element in air quality management (mitigation strategies) and air quality modelling a study, which is why it is believed to be more reliable data.

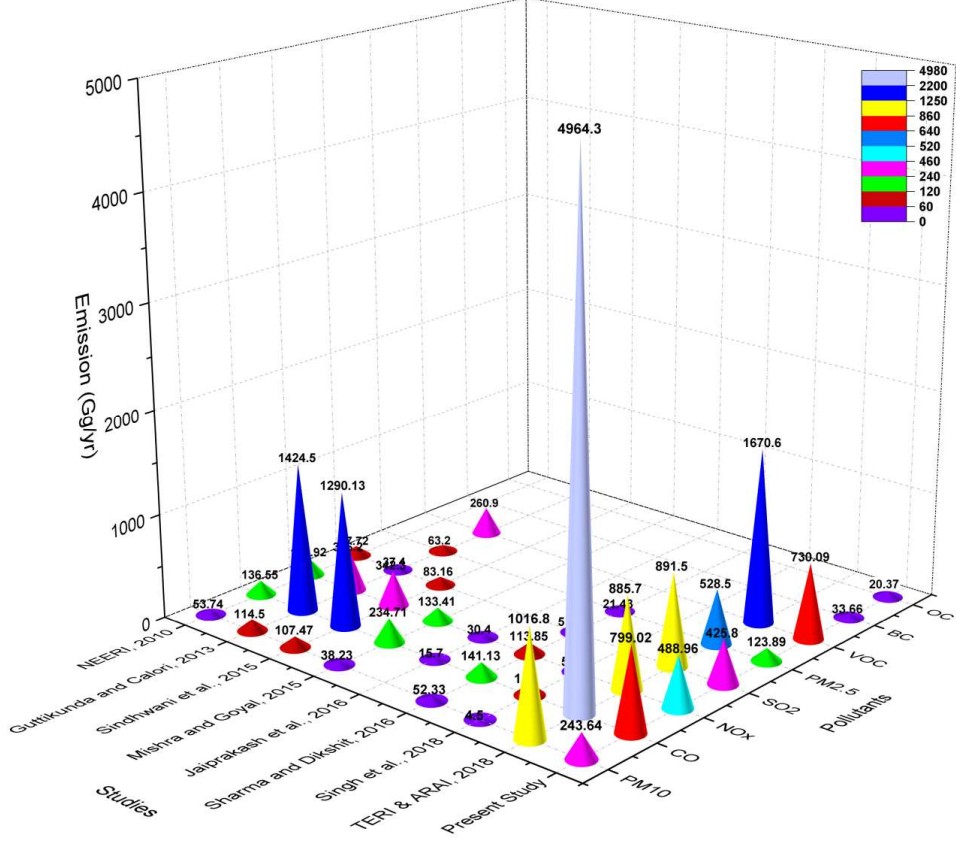

**Figure 7: Inter-comparison among studies over the domain of interest**

### 3.6.   Mitigation strategy using developed emission:

As emission inventory acts as a fundamental tool by both policymakers and scientific communities for mitigation strategies in combating air pollution in any cities. In the present



study, a developed sensitive piece of surface-gridded emission database is unique in many ways
and will pave a path to understanding the air quality issues in megacity Delhi. For the same, a
thorough analysis has been made to identify the contribution of major sectors to the high
emitting polluted zone across the Delhi. Following that, a number of hotspot regions were
identified as shown in Figure 8 (a), from which the top ten hotspots are being identified along
with first three dominating sectors affecting the air over the hotspots significantly as shown in
Figure 8 (b). Since, PM$_{10}$ is considered to be one of the dominating pollutants in modulating
urban air quality. In one of the applications to the developed emission inventories, sector-specific
control strategies are recommended based on the input of available activity data and emission
factors, which would possibly benefit the policymakers and help in the improvement of megacity
air quality. The ten most dominating hotspots are identified with the relative contribution of three
major sectors in descending order as identified in the table to follow. Each area mentioned
against each megacity below is accompanied by several color codes which denote a specific
sector associated with the pollution where; *TRN- Transport, WB- Wind-blown road dust, IND- Industry,*
*TPP- Thermal Power plant, SLM- Slum, MSW- Municipal Solid Waste burning, DG- Diesel Generator, WTE-*
*Waste-to-energy plant, RES- Residential, CON- Construction.*

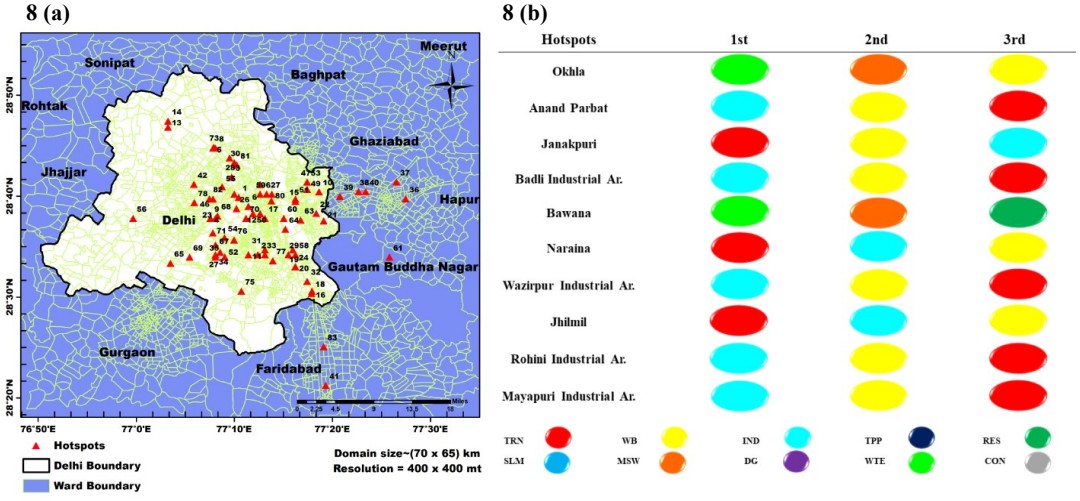


**Figure 8: (a) Hotspots across the Delhi-NCR domain, with (b) First three dominating**
**sectors affecting the air over the hotspots**



Based on the analysis of each hotspot as delineated in Figure 8, the mitigation strategy can be
framed accordingly to control the emission at source through various approaches on the ground.
Apart from this, a few sector-specific, generalized recommendations have been listed below for
all the megacities:-
a) The discard of ageing vehicles (more than 10yr) especially commercial cars and heavy
commercial vehicles category from the system followed by fast traffic movement along with
enhanced penetration of electric vehicles can reduce the transport-related emission
significantly. The heavy and light commercial (diesel) vehicles together contribute ~40-50%
of road transport emissions where strict implementation of BS-VI norms needs to be applied.
b) Vehicles from surrounding states/regions play a significant role, where the average low traffic
speed is major roads cause of elevated emission of pollutants across the megacity, so a similar
stringent vehicular policy has to be implemented in surrounding states of Delhi too.
c) Major identified roads in megacity need road diversions in order to reduce the vehicle density,
which will ultimately increase the speed of vehicles by reducing emission load from tailpipes.
d) Flexible office hours and work from home culture could be an alternative approach to reduce
traffic congestion and at the same time, will increase average speed of vehicles and associated
reduction in emissions.
e) In order to reduce the impact of silt load, Road shoulders must be repaired in regular intervals
to avoid impaired and fractured ways. Similar approach should be adopted around outskirt of
Delhi too. They should be cleaned periodically.
f) Implementation of more stringent standards for both large and small-scale industries along
with better solid/fossil fuels utilization.
g) Open burring at Municipal solid waste dumping sites should be replaced with other substitute
approaches like vermi-composting, natural decomposition, or mulching and encourage WTE
plants.
h) Slum clusters with better penetration of LPG-based cooking fuel usage to discourage solid
fuels like fuel wood, cow dung, and coal.
i) Construction sites should be properly handle materials while loading and unloading
procedures.
j) Discouraging usage of DG-set usage in unorganized industries and commercial and private
zone could potentially help reduce the emission further.




**4.      Data availability:**

The    emission    dataset   can   be   accessed   through   open   access   data   repository
https://doi.org/10.5281/zenodo.7715595 (Sahu et al., 2023).The dataset is presented in .shp file
format covering Delhi-NCR region having domain size of 70km×65 km.

**5.      Conclusion:**

Present megacities are facing pressing air quality challenges in South Asia due to variety of
individual regional sources and changing policy, therefore, the present study is attempt to decode
the understanding of present air quality over megacity Delhi through ultra-fine Emission
Inventory for 2020 proclaims to be an essential component not only to address the mitigation
plan towards improving megacity air quality but also understand the decadal change (2010-2020)
in emission patter in megacity Delhi and surrounding NCR.The decadal change with changing
government policy and action plan has modulated the emission from various unattended sources.
However, only a single strategy cannot tackle the elevated air pollution issues in Delhi-NCR. A
mixture of policy measures well adapted for domain's hotspot-specific, source-specific strategies
is imperative to improve air quality. The developed surface emission dataset provides every such
detail which can be comprehended as robust in all terms.

**Author contributions:**
Saroj Kumar Sahu (SKS) conceived the present idea and Poonam Mangaraj (PM) wrote the
whole paper and analyzed the data. Gufran Beig (GB) provided useful discussion and suggested
a conclusion.

**Acknowledgements:**
Authors acknowledge Ministry of Earth Science, Govt. of India, for partial support through
QUISARC project (Grant No. MoES/Indo-Nor/PS-10/2015). This full research work did not
receive any specific grant from funding agencies in the public, commercial, or not-for-profit
sectors



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
