# Peer review of "Decadal Growth in Emission Load of Major Air Pollutants in Delhi"

_Earth System Science Data, 2023_

## Author Response (AR1)

**(1) Comments from Referees:-**

**Referee 1:**

This study conducted extensive field surveys, and I believe that the resulting highresolution dataset is highly meaningful. I have several questions that I would like to ask:

- 1. Why did the study focus on CO and PM5in the results and discussion?
- 2. Line 113 mentions that the inventory has 17 sectors, but only 5 major sectors are introduced, which can be confusing. Could you please clarify this point?
- 3. Would you consider further developing the discussion on NH3 emissions? It would be helpful for air quality simulations.
- 4. Why did the industry OC emission vanish in 2020? And why did residential BC and OC emissions increase compared to 2010?
- 5. Could you please discuss the influence of COVID-19 on emissions in 2020?
- 6. Also, please note to use subscripts for species in the main text.

**Referee 2:**

Congratulation on the new gridded data for Delhi! A very interesting study has established a high-resolution (0.4km) emission inventory for the national capital Delhi. The best thing is that this study has undergone field surveys, which may help to improve the estimation, resulting in a high-resolution dataset that is highly significant. The paper is very well written, and the analysis is quite up to the mark where the hot spots are well identified with the proposed mitigation plan. The data generated is vital from a regional/city modeling point of view. The manuscript has minor/few grammatical mistakes to be taken care of. Following are some of the comments that could be considered:-

- 1. There are many small sources included in the study but how these sources can be mitigated with govern
- 2. Do the authors believe there could be some other sources apart from the mentioned ones, which need preliminary attention?
- 3. Do you think a total waving off the petrol and diesel vehicles and converting them to electrical mode could be a possible mitigation strategy to be considered? If "Yes" do explain how the recent e-vehicle has impacted total transport emissions and is it significant impact as of now.
- 4. Why there are many factors that lead to the large discrepancy in all reported emission estimations over Delhi and NCR? Can it be improved by adapting some approach?
- 5. How the road dust can be controlled? The author should recommend mitigation.
- 6. There is a large difference in sectorial uncertainty and how this can be improved.
- 7. Some sentences can be cut short to avoid a lengthy description at once.
- 8. If possible, the Introduction section can be reduced.
- 9. The species names should be subscripted in the texts, especially in the abstract.
- 10. In some places, proper spaces are absent between words which should be taken care of as it gets a little difficult for readers.

**(2) Author's Response:-**

**Response to Referee 1:**

- 1. Thank you for the comment. The study includes the gridded emission estimates for all the major eight pollutants i.e. PM2.5, PM10, CO, NOx, SO2, VOC, BC and OC. where the spatial patterns for all pollutants are more and less similar due to the same source of emission. However, to avoid similar discussions repeatedly for readers and space constraint, a comprehensive analysis of the spatial distribution of one particulate pollutant i.e. PM10 and one gaseous pollutant i.e. CO has been elaborated.
- 2. We have categorized all the 17 sectors under 5 major sectors which are:- (i) Transport, (ii) Windblown Road Dust, (iii) Industry, (iv) Residential (includes sub-sectors: household, slum, street vendor, crop residue burning, cow-dung, and diesel generators), (v) Others (includes sub-sectors: municipal solid waste burning, construction, incense sticks/mosquito coils/cigarettes, and crematory). Each and every sector is described and covered under all five major categories.
- 3. Ammonia (NH3) plays a vital role in atmospheric chemistry, as it is a primary form of reactive nitrogen. The sources to NH3 emission could be natural sources including agriculture, livestock excreta/manures, and animals, and anthropogenic sources including fertilizer application, crop residue, compost, vehicular exhaust, biomass burning, waste disposal and fossil fuel combustion. Therefore, certainly at present, we are in process to develop new ammonia inventory for India for recent year and a further analysis and findings will be followed separately in another paper.
- 4. Industrial OC emission has not been estimated due to unavailability of appropriate fuel-specific emission factors. Residential sector for the present study includes many new updated fuel specific/technological emission factors for many unattended sources (household cooking, slum, street vendor, burning in household, crop residue burning, cow-dung, and diesel generators) which have resulted in elevated BC and OC emissions unlike the residential sector (domestic cooking, cow-dung, crop-residue burning, slum) SAFAR-Delhi 2010 estimates.
- 5. We thank the referee for raising this question. In response to the comment we would like clarify that our surface emission data has been compiled for the period of 'April 2019 to March 2020' and this period does not include the COVID-19 pandemic and nationwide lockdown scenarios. The lockdown started towards the end of the month of March 2020 so we are optimistic that present estimates do not have an influence of the COVID-19 pandemic. Of course, there will be decrease in emission for many sectors except residential cooking and power sector.
- 6. Thank you for the positive remark about the importance of this kind of gridded surface emission data based on field survey. The same will be rectified in the text.

**Response to Referee 2:**

 Yes, Government is taking various initiatives of mitigation actions like more penetration of LPG in slums, which is seen between 2010 to 2020. Now, more than 90% slum households use LPG, which was just ~30% in 2010. Similarly, street vendors are discouraged to use biofuel and fossil fuels. Now many telecom towers are solar based power backup facilities. Construction activities are being stopped/ limited inside the city premises. Many roads are cleaned in regular interval.

- 2. We thank the referee for raising this question. We believe perhaps there might be some more unattended small source, which prevails around, but the sources, which we included here, is important in term of relative contribution, relatively higher and first of its kind 17 sectors are taken into account also. An extensive study (ground-based survey) with regard to the identification of sources region-wise or as a whole would be helpful to reach the goal.
- 3. Of course, converting the petrol and diesel vehicles to electrical mode would reduce the emission levels up to some extent but the impact may not be significant, as recent policy has encouraged electronic 2W mostly followed by 3W vehicle only. In the same line, the increase in consumption of coal in thermal power stations would on the other hand add up to the gross emissions with limited improvements. However, if the energy used in e-vehicles could be from green/renewable energy then it might some meaningful impact. Government is encouraging the e-vehicle through various cheap loan/ subsidy scheme hence could be used for running the evehicles, hence reducing the elevated emission from the traditionally dominating transport sector. With the penetration of e-car and heavy vehicle, the impact of evehicle will be meaningful.
- 4. The large discrepancy is due to varying technological emission factors used, sector specific activity data used domain size, base year and sectors. The discrepancy can be reduced by adopting standardized new technological emission factors for minor and major sectors and more of primary activities data. Many cases, the methodology adopted by various estimations is not standard and uniform. Many earlier studies adopted top-down approach due to data limitation, which is proved in present estimation by adopting bottom-up approach.
- 5. Thank you for the question. For you might have missed out, we have already mentioned possible mitigation with respect to road dust in Line no. 724. That is; "In order to reduce the impact of silt load, road shoulders must be repaired in regular intervals to avoid impaired and fractured ways. Heavy vehicles and commercial cars needs to restricted/limited inside the cities. A similar approach should be adopted around the outskirts of Delhi too. Road dust should be cleaned periodically."
- 6. The uncertainty estimation includes several factors, which are mentioned in section 3.4. in detail. Besides that, the difference in sectorial uncertainty is major because of the unavailability of country-specific and fuel/technology-specific emission factors and activity data. There are many unorganized industries in study domain where the access to fuel data is limited. If new regional emission factors are developed for each technology used in various sectors then it would have reduced the uncertainty in emission up to a larger extent. More micro-level primary activity data will help to improve the uncertainty estimations.
- 7. Thank you for the remark. We shall try to shorten a few sentences without much alteration in the text.
- 8. Thank you for the remark. We shall try to reduce the introduction section if possible.
- 9. Thank you for the remark. The same shall be rectified in the text.
- 10. Thank you for the remark. The same shall be rectified in the text.

**(3) Author's change in the manuscript:-**

- 1. In response to the comment of referee 1, we have added the detailed categorization of the sectors in Line no. 144-148.
- 2. In response to the comment of referee 1 and 2, the names of species have been subscripted at places where needed. Refer to Line no. 19, 22, and 24.
- 3. In response to the comment of referee 2, some sentences in the introduction section have been cut short to avoid a lengthy. Refer to Line no. 54, 56, 58, 62, 68, 69, 72, 73, and 74.
- 4. In response to the comment of referee 2, spaces between words and grammatical errors have been checked and rectified. Refer to Line no. 42, 52, 56, 167, 344-345, 393, 400, 464-466, 469-470, 474, 478, 486, 552, 556, 560, 575, 580, 638, 670, 683, and 717.